# Problema del Conjunto Dominante basado en la Distancia

**L. Cruz**     **E. Barrena**     **A.D. López-Sánchez**     **A.G. Hernández-Díaz**

Department of Economics, Quantitative Methods, and Economic History
Universidad Pablo de Olavide
Seville, 41013, Spain

## Abstract

El Problema del Conjunto Dominante basado en la Distancia (DDSP por sus siglas en inglés) es una variante del problema clásico de dominación en grafos en el que se consideran grafos ponderados donde los pesos de las aristas representan distancias. Se trata de un problema de optimización biobjetivo que busca tanto minimizar el tamaño del conjunto dominante como la distancia del nodo más alejado a dicho conjunto. A lo largo de este proyecto, se propondrá, analizará y resolverá el DDSP mediante un método metaheurístico. En particular, se propondrá implementar una variante de una conocida metaheurística, GRASP, y se evaluará su desempeño en distintos tipos de grafos.

## 1    Motivación

Los problemas de dominación en grafos han sido ampliamente estudiados a lo largo de las últimas décadas debido a su vasta aplicabilidad en el mundo real. Específicamente, el Problema del Conjunto Mínimo Dominante (MDSP por sus siglas en inglés) tienen como objetivo encontrar el subconjunto de nodos de menor tamaño, cumpliendo que cada nodo no perteneciente al subconjunto sea adyacente a al menos un nodo del subconjunto. La ubicación de servicios, diseño de redes de comunicaciones y análisis de redes sociales son ejemplos de aplicaciones prácticas de este problema. Sin embargo, el problema clásico de dominación no tiene en cuenta las distancias entre los nodos dominantes y sus dominados. Esto puede derivar en soluciones donde algunos nodos, a pesar de estar dominados, se encuentren a gran distancia de su nodo dominante, lo que se traduce en conexiones poco eficientes en términos de costos, tiempos o calidad.

Por ello, este problema se centra en la búsqueda de un equilibrio entre el tamaño del conjunto dominante, que interesa que sea mínimo, y la distancia del nodo más alejado al conjunto, que también se desea minimizar, mejorando de esta forma la calidad de las soluciones obtenidas.

La Figura 1 incluye un ejemplo de dos soluciones óptimas para el MDSP, con cardinalidad tres, pero, sin embargo para el DDSP, es mejor el conjunto dominante representado en la Figura 1b ya que la distancia máxima es más pequeña (15 versus 7).

Por ejemplo, en problemas de ubicación de servicios, el DDSP no solo optimiza la cantidad de instalaciones y el personal necesario, es decir, el conjunto dominante, sino que también reduce los costos operativos asociados a las distancias recorridas y los tiempos de acceso de los usuarios. En redes inalámbricas, permite disminuir el consumo energético al minimizar la distancia de transmisión entre dispositivos, mejorando así la calidad y estabilidad de las comunicaciones. En redes sociales puede optimizar la propagación de la información mediante una selección estratégica de usuario.

XVI XVI Congreso Español de Metaheurísticas, Algoritmos Evolutivos y Bioinspirados (maeb 2025).

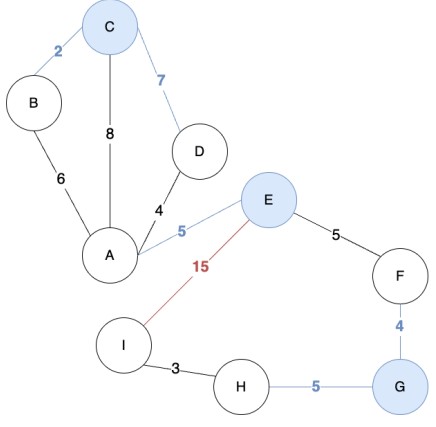
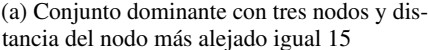
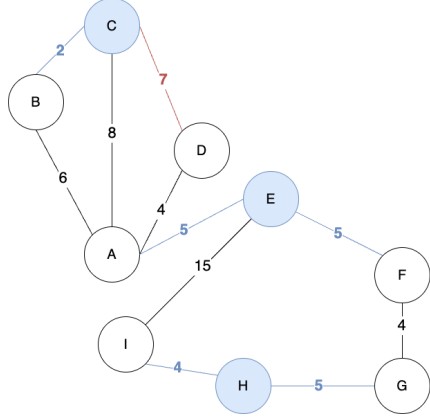

(a) Conjunto dominante con tres nodos y distancia del nodo más alejado igual 15

(b) Conjunto dominante con tres nodos y distancia del nodo más alejado igual 7

Figure 1: Ejemplo de conjuntos dominantes con igual cardinalidad pero distintas distancias máximas.

De manera general, el DDSP pretende mejorar la calidad y eficiencia de las conexiones en múltiples sistemas, acercando el problema de dominación a un plano más real.

## 2    Revisión de la literatura

El problema de dominación en grafos fue introducido por primera vez en los estudios de teoría de grafos de Ore a mediados del siglo pasado (Ore (1962)), aunque su formulación y análisis más detallados se encuentran en trabajos posteriores como (Cockayne and Hedetniemi (1977)) y (Haynes et al. (1998)).

Desde su primera aparición, múltiples líneas de investigación sobre el problema han sido exploradas gracias a su gran aplicabilidad. Se han propuesto diversas variantes como el conjunto k-dominante (Bermudo et al. (2022)), en el que cada nodo no dominante debe ser adyacente a al menos k nodos dominantes. Para el conjunto dominante total (Henning and Yeo (2014)), se exige que todos los nodos sean adyacentes a al menos un nodo dominante. En cuanto a la estructura del conjunto dominante, en la dominación conectada (Sampathkumar and Walikar (1979)), este conjunto debe inducir un subgrafo conectado, mientras que la dominación independiente (Goddard and Henning (2013)) busca conjuntos dominantes en el que sus nodos no estén conectados entre sí. Para grafos ponderados, se tienen el problema de dominación k-ponderado (Barrena et al. (2025)), en el que los pesos de las aristas representan el grado de conexión entre los nodos. En este caso, se impone la restricción de que cada nodo debe estar dominado con un grado superior al umbral k.

Respecto a la resolución de los problemas de dominación y sus variantes, debido a su complejidad $\mathcal{NP}$-difícil, el desarrollo de técnicas metaheurísticas ha cobrado gran relevancia en la literatura, ya que ofrece soluciones aproximadas en tiempos computacionales razonables. Diversos enfoques han sido propuestos para abordar el problema de dominación, como el diseño de algoritmos genéticos (Hedar and Ismail (2010)) o el desarrollo de un algoritmo iterativo (Casado et al. (2023)), basado en la destrucción y construcción de la solución. Otro método explorado ha sido el algoritmo de colonia de hormigas para el problema de la dominación conectada (Bouamama et al. (2019)). Estas estrategias han demostrado ser efectivas en la obtención de soluciones de alta calidad, especialmente en instancias de gran tamaño donde la resolución mediante métodos exactos resulta inviable.

## 3    Objetivos

Los objetivos principales del proyecto son:

- Revisar el estado del arte de los problemas de dominación en grafos y sus variantes.
- Definir formalmente y analizar el DDSP.

- Modelar matemáticamente el problema y llevar a cabo su resolución mediante un algoritmo exacto hasta detectar los límites del mismo.

- Diseñar y desarrollar un algoritmo metaheurístico basado en GRASP para la resolución del problema haciendo uso de las características topológicas o estructurales de los grafos.

- Validar el desempeño del algoritmo metaheurístico implementado con los resultados obtenidos por un algoritmo exacto.

- Comparar experimentalmente el algoritmo metaheurístico diseñado con los algoritmos del estado del arte.

- Analizar e interpretar los resultados obtenidos por el nuevo algoritmo.

## 4  Hipótesis

### 4.1  Definición del problema

Sea $G = (V, E, d)$ un grafo ponderado no dirigido con $V$ el conjunto de nodos, $E \subseteq V \times V$ el conjunto de aristas y $d : E \to \mathcal{R}^+$ una función que asigna valores positivos a cada arista $(u, v) \in E$.

**Conjunto dominante**   Se dice que un conjunto de nodos es un conjunto dominante si todo nodo del grafo o pertenece al conjunto dominante o es adyacente a al menos un nodo del conjunto dominante.

Formalmente, un subconjunto $D \subseteq V$ es un conjunto dominante si:

$$\forall u \in V \setminus D, \exists v \in D \text{ tal que } (u, v) \in E.$$

**Distancia de un nodo a un conjunto**   Se define la distancia de un nodo a un conjunto de nodos como la menor distancia entre dicho nodo y cualquier nodo del conjunto, siendo este valor igual a cero si el nodo pertenece al conjunto.

Formalmente, la distancia de un nodo $u \in V$ a un conjunto de nodos $D \subseteq V$:

$$d(u, D) = \min\{d(u, v) : v \in D\}$$

Luego, el DDSP puede definirse de la siguiente manera:

$$\min_{D \subseteq V} \left( |D|, \max_{u \in V \setminus D} d(u, D) \right), \quad \text{sujeto a que } \forall u \in V \setminus D, \exists v \in D \text{ tal que } (u, v) \in E.$$

### 4.2  Resolución del problema

El DDSP es un problema de optimización biobjetivo cuya resolución se aborda en este proyecto utilizando el método de las $\varepsilon$-restricciones. Este enfoque convierte el problema biobjetivo en uno monobjetivo al incluir uno de los objetivos como una restricción, acotando su valor por un umbral $\varepsilon$. Al resolver el problema para distintos valores de $\varepsilon$, se obtiene una curva de Pareto.

El $\varepsilon - DDSP$ consiste en encontrar el menor conjunto dominante del grafo de forma que la distancia del nodo más alejado al conjunto dominante sea menor que el umbral $\varepsilon$.

Formalmente, dado un valor $\varepsilon \in \mathcal{R}^+$, el $\varepsilon - DDSP$ se define de la siguiente forma:

$$\min_{D \subseteq V} |D|, \quad \text{sujeto a que } \forall u \in V \setminus D, \exists v \in D \text{ tal que } (u, v) \in E \quad \text{y} \max_{u \in V \setminus D} d(u, D) \leq \varepsilon.$$

A lo largo del proyecto se resolverá el $\varepsilon - DDSP$ con un algoritmo exacto a fin de detectar los límites del mismo y, posteriormente, se pasará al diseño e implementación de un algoritmo metaheurístico basado en GRASP para la obtención de soluciones de buena calidad en un tiempo computacional razonable. El estudio y análisis en detalle del problema, así como de las características de los grafos, permitirán la inclusión de conocimiento al algoritmo con el objetivo de mejorar su desempeño.

## 5 Descripción de la metodología

**GRASP (Greedy Randomized Adaptive Search Procedure)**    Metaheurística de tipo trayectorial y multiarranque, desarrollada a finales del siglo XX (Feo and Resende (1989); Feo et al. (1994)) y ampliamente utilizada para la resolución de problemas de optimización combinatorios. Su principal característica es la combinación de estrategias de intensificación y diversificación en dos fases bien diferenciadas: la fase de construcción y la fase de mejora local.

GRASP consta de $N$ iteraciones en las que se ejecutan de manera secuencial cada una de las fases previamente mencionadas. La fase de construcción genera un solución inicial mediante un proceso iterativo en el que, partiendo de la solución vacía, se van insertando elementos hasta conseguir una solución factible. La selección de estos elementos, aunque sigue un criterio inteligente, incorpora un cierto grado de aleatoriedad a fin de explorar diferentes regiones del espacio de soluciones, favoreciendo de esta forma la diversificación. La fase de mejora local tiene como punto de partida la solución inicial previamente generada y se le aplica un procedimiento de búsqueda local. Este proceso se basa en realizar pequeñas modificaciones incrementales a la solución con el objetivo de optimizarla. La búsqueda local impulsa la intensificación, permitiendo alcanzar soluciones de mayor calidad dentro del vecindario explorado. Este proceso se repite $N$ veces, obteniendo así $N$ soluciones. Finalmente, se selecciona la mejor de ellas.

El objetivo del proyecto es diseñar una metaheurística basada en GRASP para la resolución del DDSP. Para ello, se implementarán cada una de las fases del algoritmo: construcción y búsqueda local. El diseño de cada etapa tendrá en cuenta las particularidades del problema gracias a la incorporación de conocimiento específico de este.

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
