# OpenReview forum: "Problema del Conjunto Dominante basado en la Distancia"
_MAEB/2025/Projects_Track — MAEB 2025 Proyectos_

### Official Review · Reviewer_6xfX · 2025-03-17
**Problema del Conjunto Dominante basado en la Distancia**

**Rating:** 5
**Confidence:** 4

**Review:**

Este proyecto propone el diseño y desarrollo de la metaheurística GRASP para resolver el Problema del Conjunto Dominante basado en la Distancia. Se trata de un problema bi-objetivo, donde el objetivo es minimizar el número de nodos en el conjunto dominante y a su vez minimizar la distancia del nodo más alejado a dicho conjunto.

En vez de utilizar un algoritmo multi-objetivo, se propone realizar un enfoque con un único objetivo, minimizar el tamaño del conjunto dominante, siendo la distancia del nodo más alejado una restricción a cumplir.

El proyecto está bien planteado, los objetivos son sensatos, y particularmente me parece interesante realizar una primera aproximación basada en un algoritmo exacto. No obstante, y tal y como se indica, DDSP parece estar directamente relacionado con “Antenna placement problem” y similares, por lo que es importante realizar una búsqueda de este tipo de problemas de optimización e identificar los algoritmos propuestos, que imagino serán abundantes. En este sentido, no sé cual ha sido el motivo que ha llevado a las/os proponentes del proyecto a centrarse en GRASP, pero mantendría una puerta abierta hacia otras opciones en función de los resultados de la revisión bibliográfica.

Finalmente, puede que no tenga sentido, pero el problema me recuerda en cierto modo al problema de clustering, y me pregunto si los algoritmos diseñados, especialmente aquellos que tienen en cuenta datos ingentes (big data), podrían ser una opción a estudiar.

---

### Official Review · Reviewer_nNyB · 2025-03-19
**Los autores pretenden resolver el problema del conjunto dominante basado en distancia mediante mediante una metaheurística denominada GRASP y realizar las comparativas con respecto a un método exacto. Este problema tiene interesantes aplicaciones prácticas. En general está bien redactado y es bastante interesante. Sin embargo requeriría cierta contextualización y aclaratoria de algunos puntos importantes que se describen en la revisión.**

**Rating:** 2
**Confidence:** 3

**Review:**

Los autores pretenden resolver el problema del conjunto dominante basado en distancia mediante mediante una metaheurística denominada GRASP y realizar las comparativas con respecto a un método exacto. Este problema tiene interesantes aplicaciones prácticas. En general está bien redactado y es bastante interesante.

Comentarios

El objetivo número 3 va a depender específicamente de las instancias del problema, y por lo tanto debería definirse un límite acerca de esas instancias; en este contexto imagino que pretenden buscar el límite superior e inferior para dichas instancias. En cuanto al objetivo 4 no veo cómo justifican el uso de grasp por encima de otro método metaheurístico. Generalmente, no hay un método único para la solución de problemas de este estilo mediante metaurísticas, por lo que a veces se requiere el uso de dos o más metaheurísticas y realizar el análisis experimental correspondiente para determinar el mejor método para resolver el problema deseado.

Por otra parte, se menciona que el problema es biobjetivo, sin embargo se está simplificando a un problema monoobjetivo, la pregunta fundamental aquí sería por qué no utilizar técnicas directamente multiobjetivo. Sería interesante que los autores  justificaran este elemento de diseño.

Por otra parte, creo que habría que ajustar el título porque es demasiado genérico.

---

### Decision · Program_Chairs · 2025-03-20

Accept